# The ER morphology-regulating lunapark protein induces the formation of stacked bilayer discs

Songyu Wang[1], Robert E Powers[1], Vicki AM Gold[2,3,4], Tom A Rapoport[1]

**Lunapark (Lnp) is a conserved membrane protein that localizes to and stabilizes three-way junctions of the tubular ER network. In higher eukaryotes, phosphorylation of Lnp may contribute to the conversion of the ER from tubules to sheets during mitosis. Here, we report on the reconstitution of purified Lnp with phospholipids. Surprisingly, Lnp induces the formation of stacked membrane discs. Each disc is a bicelle, with Lnp sitting in the bilayer facing both directions. The interaction between bicelles is mediated by the cytosolic domains of Lnp, resulting in a constant distance between the discs. A phosphomimetic Lnp mutant shows reduced bicelle stacking. Based on these results, we propose that Lnp tethers ER membranes in vivo in a cell cycle–dependent manner. Lnp appears to be the first membrane protein that induces the formation of stacked bicelles.**

## Introduction

A major feature of the ER is a network of tubules interconnected by three-way junctions. The network is shaped by several membrane proteins (1, 2, 3, 4, 5). One class of proteins comprises curvature-stabilizing proteins, including members of the reticulon and receptor expression–enhancing protein families (6, 7). The other important class consists of membrane-fusing GTPases, including the atlastins (ATLs) in metazoans and Sey1p and their homologs in yeast and plants (8, 9, 10, 11). Recent reconstitution experiments showed that a GTP-dependent network can be generated with proteoliposomes containing one member of each protein class (12), indicating that a curvature-stabilizing and a membrane-fusion protein are the minimal components required for ER network formation and maintenance. However, these results do not exclude that other proteins play a role in network formation. In fact, the lunapark (Lnp) protein has been proposed to be an additional player in shaping the tubular ER network (13, 14, 15, 16).

Lnp is an ER membrane protein found in all eukaryotic cells. The protein localizes preferentially to three-way junctions of ER tubules and has been proposed to stabilize the junctions (14, 15, 16). However, reconstitution experiments showed that it is not required for three-way junction formation (12). Deletion of Lnp in mammalian tissue culture cells also does not abolish the tubular ER network, although tubules and three-way junctions become less abundant (16). The inactivation of Lnp in interphase *Xenopus laevis* egg extracts converts three-way tubular junctions into small sheets (16). This morphological transition is similar to that observed in mitotic extracts (16), suggesting that inactivation of Lnp during mitosis contributes to the known conversion of the ER from tubules to sheets during mitosis (17). Indeed, both *Xenopus* and human Lnp are phosphorylated at several sites during mitosis (16). However, how exactly Lnp affects the ER network remains unclear.

Lnp contains two closely spaced transmembrane (TM) segments, flanked by cytosolic coiled-coil domains (CC1 and CC2). In higher organisms, CC2 is followed by a phosphorylation (P) domain that contains the mitotic phosphorylation sites, a $Zn^{2+}$-finger domain, and a poorly conserved C-terminal domain that is predicted to be disordered (Fig 1A). At the N-terminus is a myristoylation site, the mutation of which disturbs the localization of Lnp to three-way junctions (16, 18). CC1, CC2, and the $Zn^{2+}$-finger are also all required for the correct targeting of Lnp to junctions (16). The $Zn^{2+}$-finger domain is involved in the dimerization of Lnp molecules (16, 19), but the role of Lnp dimers remains unknown.

Here, we report on the purification and reconstitution of Lnp. Surprisingly, we found that purified *Xenopus* or human Lnp induces very unusual structures: stacked bicelles. To our knowledge, this is the first example of a membrane protein that can induce the formation of these structures. Using various mutants of purified Lnp, we provide evidence that Lnp molecules interact across bicelles through separate sites in their cytosolic domains. Based on these results, we discuss how the biological function of Lnp is determined by cytosolic interactions between Lnp molecules sitting in different membranes.

## Results

### Purified Lnp forms stacked discs upon reconstitution with phospholipids

We first purified Lnp from *Xenopus*. The protein was expressed in *Escherichia coli* with a C-terminal His10 tag and purified in

[1]Howard Hughes Medical Institute and Department of Cell Biology, Harvard Medical School, Boston, MA, USA  [2]Department of Structural Biology, Max Planck Institute of Biophysics, Frankfurt am Main, Germany  [3]Living Systems Institute, University of Exeter, Exeter, UK  [4]College of Life and Environmental Sciences, Geoffrey Pope, University of Exeter, Exeter, UK

Correspondence: tom_rapoport@hms.harvard.edu

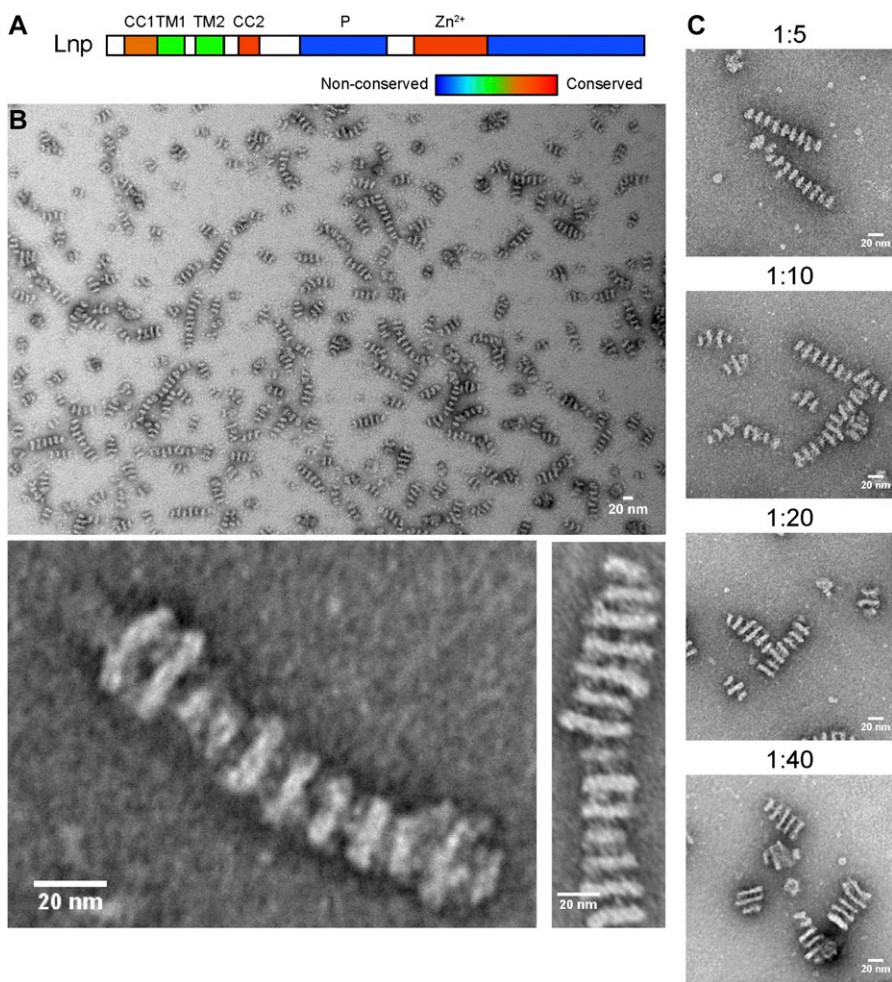

**Figure 1. Reconstituted *Xenopus* Lnp forms stacked membrane discs.**
**(A)** Domain organization of Lnp. CC1, CC2, coiled-coil domains 1 and 2; TM1 and TM2, transmembrane segments 1 and 2; P, phosphorylation domain; $Zn^{2+}$, $Zn^{2+}$-finger domain. Domains are colored according to the degree of sequence conservation. **(B)** Purified His10-tagged *Xenopus* Lnp was reconstituted with phospholipids at a protein-to-lipid molar ratio of 1:5 and visualized by negative-stain EM. The lower panel shows a magnified view. Scale bar, 20 nm. **(C)** As in (B), but with different protein-to-lipid ratios. Scale bar, 20 nm.

dodecylmaltoside (DDM) using a Ni-nitrilotriacetic acid (NTA) resin, followed by size-exclusion chromatography (SEC). The peak fractions were pooled and contained a major band of the expected size (Fig S1A and C). The purified protein was mixed with liposomes containing phosphatidylcholine, phosphatidylethanolamine, and phosphatidylserine at a ratio resembling that of the ER, and the detergent was removed by incubation with Bio-Beads. The sample was then analyzed by negative-stain EM. Essentially, all visible particles consisted of a series of stacked densities, which are likely to be discs (Fig 1B). The discs have a uniform distance from each other (12 nm), but their diameters vary even within a stack (from 13 to 22 nm). The thickness of the discs is about 5 nm. Each disc consists of two bright areas and a darker one in between, likely due to differences in accessibility of the stain (Fig 1B, magnified views). Thin threads are often seen between discs, suggesting that protein bridges are responsible for the equidistant stacking of the discs.

Stacked discs were formed over a wide range of protein-to-lipid ratios (Fig 1C). The distance between the discs remained the same, but the average diameter of the discs increased and fewer discs were assembled into each stack at a higher lipid concentration. The disc diameter varied greatly, from <15 to >50 nm.

With all protein-to-lipid ratios, reconstituted Lnp floated in a Nycodenz gradient, indicating that the protein was in fact present in lipid-containing structures (Fig S2A). This is supported by the fact that no discs were observed when purified Lnp alone was visualized at high protein concentrations in the absence of added lipids (Fig S2B). Furthermore, the structures disappeared when detergent was added after their formation (Fig S2C). Stacked discs were also seen when the His10 tag was removed from purified *Xenopus* Lnp before its reconstitution with lipids (Fig S3).

When antibodies to the cytosolic domain of *Xenopus* Lnp were added after reconstitution, the structures disappeared (Fig S4B versus A), indicating that the cytosolic domain of Lnp is required for the maintenance of the stacked disc structures. No effect was observed when the antibodies were presaturated with the purified cytosolic domain of Lnp before addition to preformed disc structures (Fig S4C), or when antibodies to other proteins were added (Fig S4D and E).

To test whether the formation of stacked discs is a general property of Lnp, we purified human Lnp after its expression as a His10-tagged protein in *E. coli* (Fig S1B and C). Again, stacked discs were observed after reconstitution at different protein-to-lipid

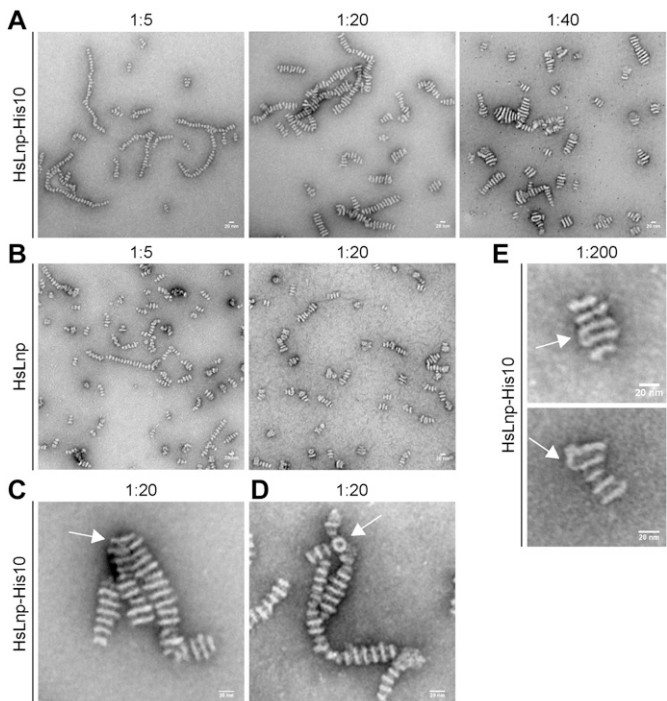

**Figure 2. Reconstituted human Lnp forms stacked discs.**
**(A)** Human Lnp was expressed in *E. coli* as a His10-tagged protein, purified, and reconstituted with phospholipids at different protein-to-lipid ratios. The samples were visualized by negative-stain EM. Scale bar, 20 nm. **(B)** As in (A), but with untagged human Lnp expressed in, and purified from, *S. cerevisiae.* **(C)** As in (A) with His-tagged protein reconstituted with lipid at a ratio of 1:20. Shown is a magnified view of a branch in which one disc is connecting two stacks (arrow). **(D)** As in (C), but with a branch point containing a ring-like structure (arrow). **(E)** As in (A) with a protein-to-lipid ratio of 1:200, showing two examples in which neighboring discs are laterally connected (arrows).

ratios (Fig 2A). Similar results were obtained when untagged human Lnp was reconstituted (Fig 2B). In this case, the protein was expressed as a streptavidin-binding peptide (SBP)–tagged protein in *Saccharomyces cerevisiae*, purified in Triton X-100 on a streptavidin column, and, after removal of the SBP tag, further purified by SEC. Despite all of these differences, human Lnp gave essentially the same stacked disc structures as the *Xenopus* protein, with a constant distance between the discs (12 nm) and a variable disc diameter. Occasionally, the structures showed branch points. One type of branching occurred when one or more discs were constituents of two stacked disc structures (Fig 2C). Another type contained a ring-like structure from which one or more stacked structures emerged (Fig 2D).

At the lowest protein-to-lipid ratio (1:200), the stacks contained fewer discs (Fig 2E). At these conditions, the discs were sometimes connected at the edges, suggesting that their stacking may be caused by the membranes folding back onto themselves, followed by the breakage of the lateral connections.

## Electron cryo-tomography of stacks

To analyze the stacked structures in more detail, samples containing reconstituted human Lnp were plunge-frozen and analyzed by electron cryo-tomography (cryo-ET). Chains of stacked dense material were visible in the reconstructed tomograms (Fig 3A), indicating that they are not an artifact of negative staining. To obtain 3D views, subtomogram averaging was performed. This showed that each layer in the stack is a disc with low density in its interior (Fig 3B–D). The distance between two neighboring layers was again 5 nm (Fig 3D), characteristic of a lipid bilayer. The low contrast in the middle of the bilayer (Fig 3D) is likely caused by the hydrocarbon chains of the phospholipids, which scatter electrons less strongly than the phospholipid head groups. These results suggest that the discs are in fact bicelles, that is, discs consisting of a single lipid bilayer with the edges of the bilayer discs forming monolayers (see scheme in Fig 3F).

As in the negative-stain images, the distance between the discs was 12 nm (Fig 3E). Between the discs is a layer of lower density that can be attributed to the cytosolic segments of Lnp. These results suggest that Lnp sits in the bilayer facing out on both sides of each disc and that the cytosolic domains mediate the interaction between the discs (scheme in Fig 3F). This model explains why the discs are separated by a constant distance.

## Lnp domains involved in stacked bicelle formation

To identify the domains of Lnp required for the formation of stacked bicelles, we first generated dominant-negative Lnp fragments that would interfere with stacked bicelle formation. To this end, we purified the cytosolic domain of *Xenopus* Lnp (cytLnp) as a His6-tagged protein from *E. coli*. When cytLnp was mixed with full-length Lnp before reconstitution with lipids, no disc structures were observed using negative-stain EM (Fig 4A). These results are consistent with previous observations that cytLnp can interact with full-length Lnp (16). The results support a model in which interactions between the cytosolic domains of Lnp molecules located in different bicelles are required for disc stacking (scheme in Fig 3F). When added after reconstitution, cytLnp had a more moderate effect (Fig 4B), suggesting that the *trans*-interactions between Lnp molecules are relatively stable once they are formed.

To analyze which subdomains of cytLnp are required for bicelle stacking, we purified various subfragments of *Xenopus* cytLnp. As with cytLnp, a dominant-negative effect was observed with a C-terminal fragment that contains the $Zn^{2+}$-finger domain and C-terminal tail (Fig 4C). On the other hand, a cytosolic domain comprising CC2 and a P domain did not interfere with stacked bicelle formation (Fig 4D). The P domain alone also had no effect (Fig 4E). The same Lnp fragments that prevent bicelle formation also act as dominant-negative reagents in *Xenopus* egg extracts, inactivating endogenous Lnp and thereby converting three-way junctions into larger sheets (16).

Next, we identified important domains in Lnp by generating deletions in the full-length *Xenopus* protein. Previous experiments with mammalian cells had hinted at an important role for the coiled-coil regions in localizing Lnp to three-way junctions (16). We first purified a mutant version that contains an insertion after TM2 and is expected to disrupt the interaction between CC1 and CC2. Consistent with the fact that this mutant protein does not localize to three-way junctions in mammalian tissue culture cells (16), it did not form stacked bicelles after reconstitution (Fig 5A; and see scheme in Fig 3F). Two Lnp

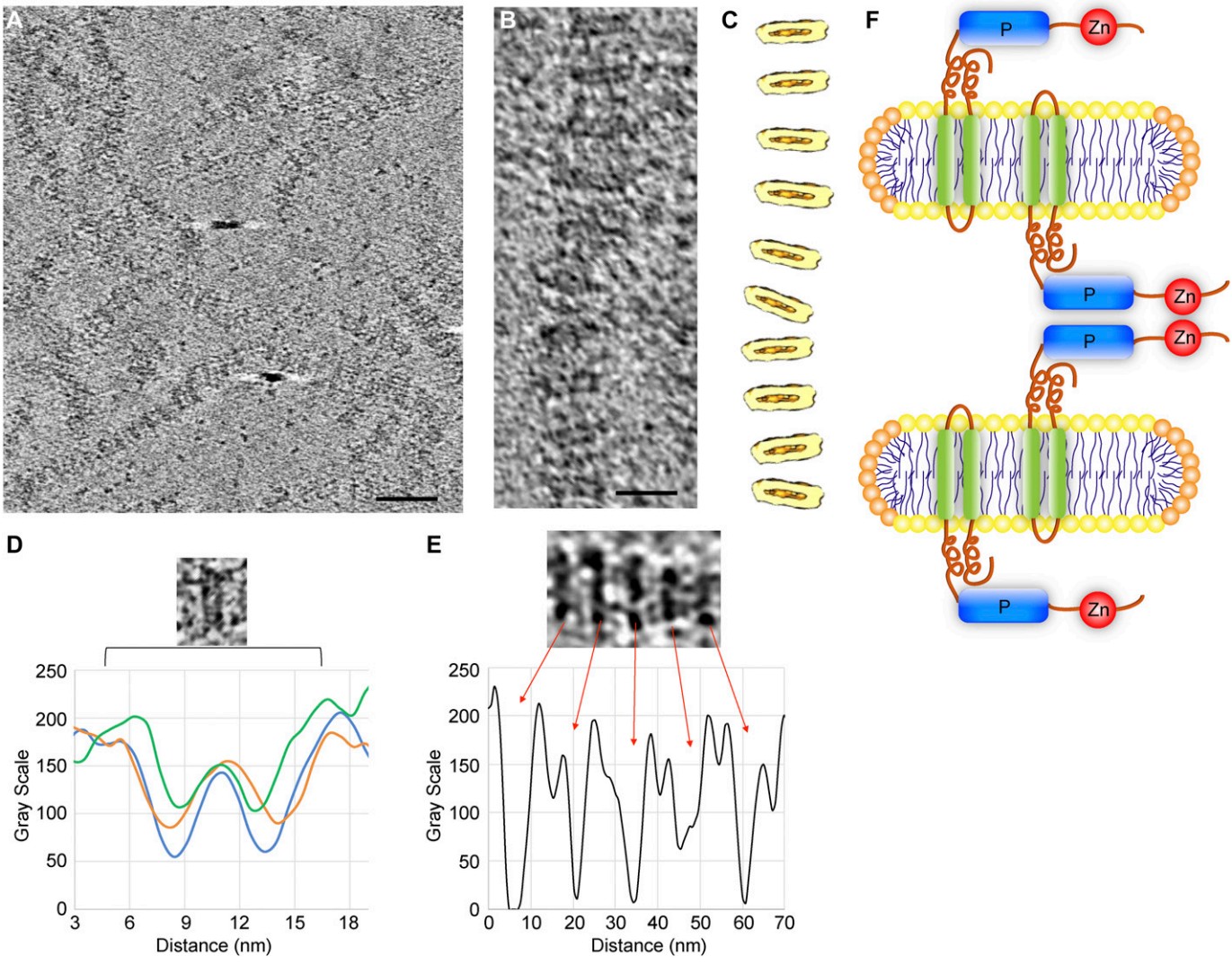

**Figure 3. Cryo-ET of stacks.**
**(A)** Human Lnp was reconstituted with phospholipids at a 1:5 molar ratio. The sample was analyzed by cryo-ET at −3 μm defocus. Shown is a slice through the tomogram. Scale bar, 50 nm. **(B)** As in (A), but at a higher zoom level. Scale bar, 20 nm. **(C)** 3D rendering of the stack shown in (B), generated by subtomogram averaging of 115 discs. The discs are shown in light yellow and the central low-density region in dark yellow. **(D)** Plot of the density profile of three individual discs (in orange, blue, and green). The distance between the high-density regions of a disc is about 5 nm. **(E)** As in (A), but with human Lnp reconstituted at a 1:20 protein-to-lipid ratio and with images taken at −5 μm defocus. The plot of the density profile shows that the discs are separated by 12 nm and are connected by low-density material. **(F)** Model of Lnp-induced bicelles, with the lipid bilayer in yellow and the detergent monolayer at the edge in orange. Lnp molecules sit in each membrane disc facing opposite directions. They interact through the P and $Zn^{2+}$-finger domains (Figs 4, 5, 6, and 7).

mutants carrying several amino acid changes in CC1 and CC2 were also inactive (Fig 5B and C). Interestingly, a mutant that lacked most of the $Zn^{2+}$-finger domain still formed stacked bicelle structures (Fig 6A), even though the $Zn^{2+}$-finger domain has been shown to be involved in the dimerization of Lnp (19). An Lnp version that lacked the entire C-terminal segment following the P domain also formed stacked bicelles (Fig 6B), although the protein was poorly behaved and formed abnormal discs. The mutant lacking the $Zn^{2+}$-finger domain could still interact with full-length Lnp, as shown by pull-down experiments (Fig 6C). Thus, the N-terminal part of cytLnp that includes the CC2 and P domains seems to mediate the stacking between bicelles (see scheme in Fig 3F). Indeed, the P domain plays an important role, as human Lnp lacking this domain generated far fewer stacked bicelles (Fig 7A; and quantification in Fig 7C). Furthermore,

introducing phosphomimetic mutations into the mitotic phosphorylation sites of the P domain of *Xenopus* Lnp (16) also impaired stacked disc formation (Fig 7B and C). Taken together, these results indicate that multiple interactions between the cytosolic domains of Lnp are required for the formation of stacked bicelles (Fig 3F). They further suggest that *trans*-interactions between Lnp molecules are regulated by mitotic phosphorylation.

## Discussion

Here, we report the surprising observation that the reconstitution of purified Lnp protein with lipids leads to the formation of stacked

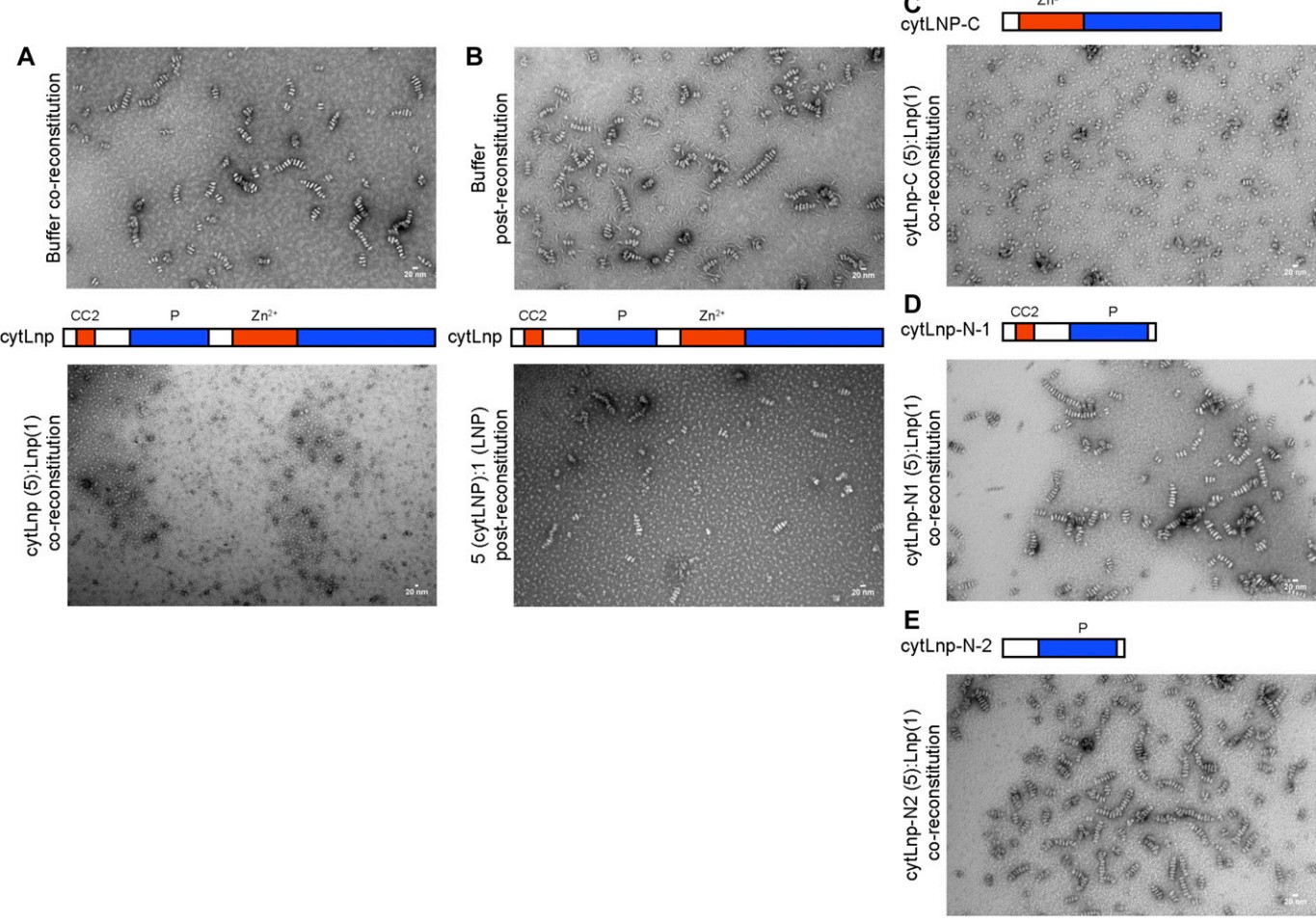

**Figure 4. Interference of Lnp domains with the formation of stacked bicelles.**
**(A)** Purified full-length *Xenopus* Lnp was mixed with phospholipids and a five-fold molar excess of purified cytoplasmic domain of Lnp (cytLnp) or buffer. After removal of detergent with Bio-Beads, the sample was visualized by negative-stain EM. Scale bar, 20 nm. **(B)** As in (A), except that cytLnp was added after reconstitution of full-length Lnp. **(C)** As in (A), except that cytLnp was replaced by a C-terminal fragment containing the $Zn^{2+}$-finger. **(D)** As in (A), except that cytLnp was replaced by an N-terminal fragment containing the CC2 and P domains. **(E)** As in (A), except that cytLnp was replaced by an N-terminal fragment containing only the P domain.

bicelles. Each of these bicelles is a membrane disc in which Lnp sits in the bilayer facing out in opposite directions. The discs are stacked with constant distances by interactions between the cytosolic domains of Lnp. These interactions involve several regions of Lnp, including the coiled-coil and P domains.

Normally, when a detergent-solubilized membrane protein is mixed with lipids and the detergent is subsequently removed, vesicles are formed in which the protein sits in the lipid bilayer in both orientations, although often one orientation is favored. Membrane proteins can also be reconstituted into bicelles when lipids and certain amphiphiles are present. This method has been successfully for the crystallization of certain membrane proteins (20). The amphiphile is required to form the monolayer edges of the membrane discs, the hydrophobic surface of which would otherwise unfavorably be exposed to an aqueous environment.

The amphiphiles used for bicelle crystallization are detergents that prefer a monolayer over a bilayer, such as dihexanoylphosphatidylcholine or 3-([3-cholamidopropyl] dimethylammonio)-2-hydroxy-1-propanesulfonate. Our Lnp bicelles were formed with proteins purified in DDM or Triton X-100, detergents that do not normally favor bicelle formation. However, the strong stacking interaction between the bicelles may have prevented the complete removal of detergent and forced the residual detergent molecules into the monolayer edges. This model would be consistent with the observation that the disc diameter increases with higher lipid concentrations (Figs 1C and 2A and B) and with the fact that DDM forms ellipsoid micelles and can sit at the edges of bicelle-like structures (21). Strong interactions between the cytosolic domains of Lnp could also explain why bicelles, rather than vesicles, are formed. Vesicles may form initially during detergent removal, but *trans*-interactions of Lnp molecules would lead to their flattening and stacking; eventually, the lateral bilayer connections would break because of the excessively high membrane curvature and then be replaced by detergent. The stacking of many small discs would be favored over that of a few, large membrane sheets because the number of Lnp molecules without a *trans*-interaction partner would be minimized (only the discs on each side of a stack contain unpaired Lnp molecules). Ultimately, the number and size

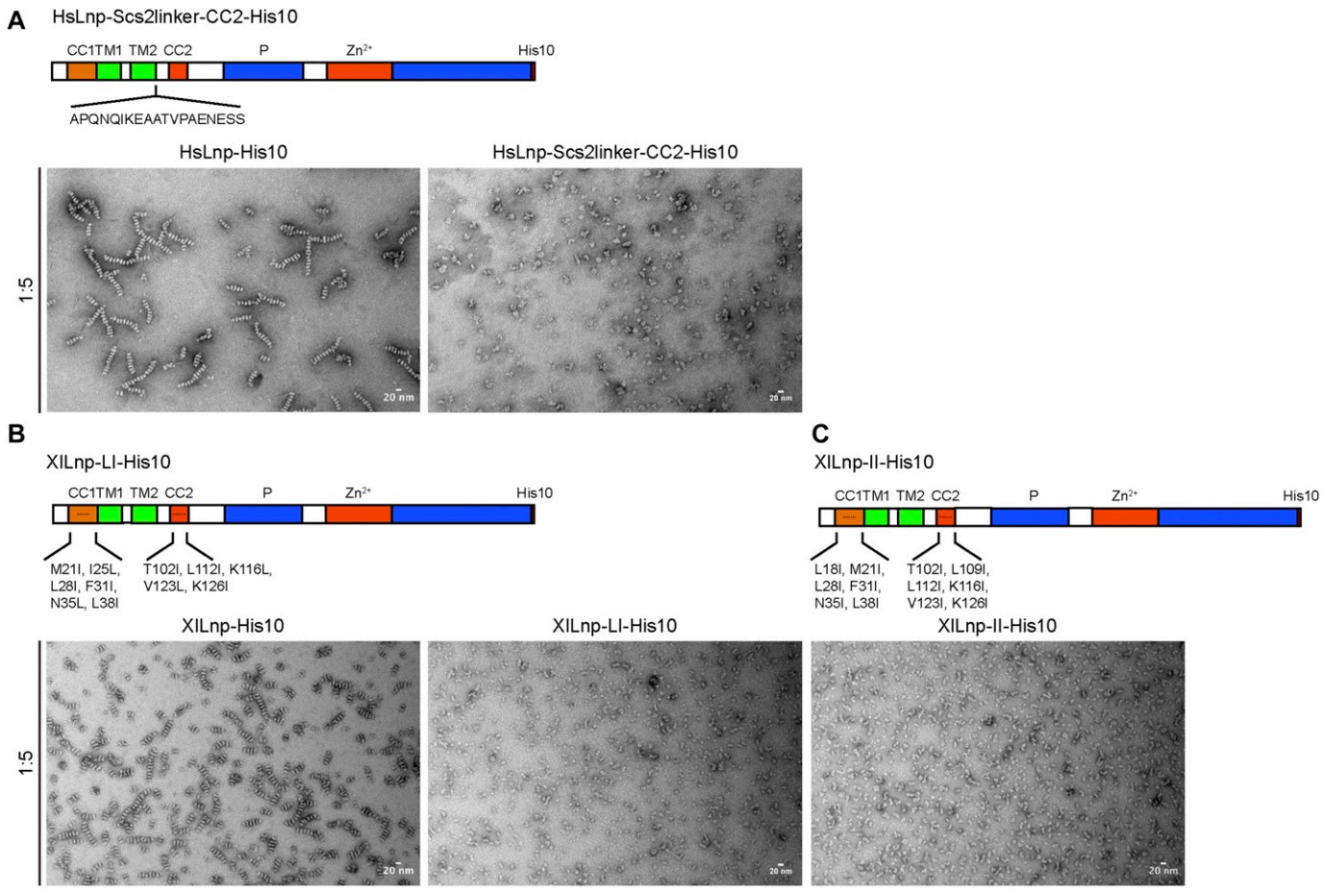

**Figure 5. The coiled-coil domains of Lnp are required for bicelle stacking.**
**(A)** Purified full-length human Lnp or an Lnp mutant carrying an insertion after TM2 was reconstituted with phospholipids at a 1:5 ratio. The samples were visualized by negative-stain EM. Scale bar, 20 nm. **(B)** As in (A), but with either wild-type *Xenopus* Lnp or a mutant that contains several point mutations in CC1 and CC2 (M21I, I25L, L28I, F31I, N35L, L38I, T102I, L112I, K116L, V123L, and K126I). The control on the left shows the same image as in Fig 1B. **(C)** As in (B), but with a mutant that contains different mutations in CC1 and CC2 (L18I, M21I, L28I, F31I, N35I, L38I, T102I, L109I, L112I, K116I, V123I, and K126I).

of the discs in a stack are determined by the interaction energy between Lnp molecules, as well as the amount of lipid and residual detergent. Although we favor the idea that the edges of the stacked bicelles are formed by detergent molecules, we cannot exclude that they are stabilized by Lnp molecules.

Lnp may be the first example of a membrane protein that induces the formation of stacked bicelles, but there are previous reports in which similar structures have been observed. For example, diacylglycerol kinase incorporated into nanodiscs forms stacked membrane discs (22); in this case, the edges of the bilayer are surrounded by scaffolding proteins that are derived from apolipoprotein AI, the main constituent of high-density lipoproteins. Interactions between the hydrophilic regions of diacylglycerol kinase might be responsible for the stacking of the discs. Reconstituted apolipoprotein AI alone can also form stacked disc structures, but it remains unclear how the discs are stacked, as the protein is thought to localize to the edges of the discs (23). Bicelles containing cationic lipids can also be stacked by the addition of DNA (24), and reconstituted apolipoprotein E4 forms artificial stacked discs when phosphotungstate is used as

negative stain, which probably serves as a linker between lipid discs (25, 26, 27).

In our system, the stacking of neighboring bicelles is mediated by *trans*-interactions between the cytosolic domains of Lnp. The coiled-coil domains seem to be one contributor to these interactions. CC1 and CC2 immediately flank the two TM segments and likely form interacting helices that are cytosolic extensions of the membrane-embedded helices of the TMs. Both CC1 and CC2 are relatively short (~20 amino acids), so in a helical conformation they would not be able to bridge the distance between the discs (12 nm) and interact in *trans* with molecules in a neighboring disc. Rather, CC1 and CC2 likely interact in *cis* in the same membrane. However, these domains probably position the P and $Zn^{2+}$-finger domains for *trans*-interactions. Our results with a deletion mutant demonstrate that the P domain is indeed required for disc stacking. The P domains alone seem to have only a low affinity for one another, as the isolated domain does not have a dominant-negative effect when added prior to reconstitution. Introducing phosphomimetic mutations into the P domain inhibits disc stacking, supporting the idea that P domains are involved in

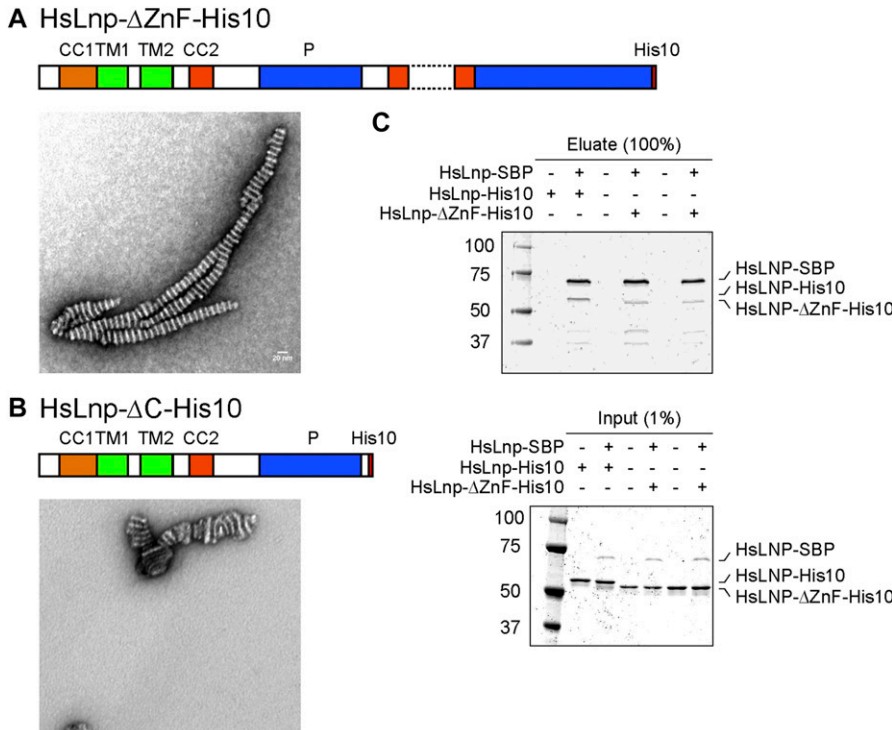

**A HsLnp-ΔZnF-His10**

CC1 TM1 TM2 CC2 P His10

**B HsLnp-ΔC-His10**

CC1 TM1 TM2 CC2 P His10

**C**

Eluate (100%)

| HsLnp-SBP | - | + | - | + | - | + |
| HsLnp-His10 | + | + | - | - | - | - |
| HsLnp-ΔZnF-His10 | - | - | - | + | - | + |

100
75
50
37

HsLNP-SBP
HsLNP-His10
HsLNP-ΔZnF-His10

Input (1%)

| HsLnp-SBP | - | + | - | + | - | + |
| HsLnp-His10 | + | + | - | - | - | - |
| HsLnp-ΔZnF-His10 | - | - | + | - | + |

100
75
50
37

HsLNP-SBP
HsLNP-His10
HsLNP-ΔZnF-His10

**Figure 6. The Zn²⁺-finger domain is not essential for the formation of stacked bicelles.**
**(A)** Purified Lnp lacking the $Zn^{2+}$-finger domain (ΔI275–F302) was reconstituted with phospholipids at a 1:5 protein-to-lipid ratio and the sample was analyzed by negative-stain EM. Scale bar, 20 nm. **(B)** As in (A), but with Lnp lacking both the $Zn^{2+}$-finger and C-terminal regions (ΔM235–E428). **(C)** Purified SBP-tagged full-length human Lnp was incubated with either His10-tagged full-length human Lnp or Lnp lacking the $Zn^{2+}$-finger domain at a molar ratio of 1:3 or 1:5. The samples were incubated with streptavidin resin, washed, and eluted with biotin. They were analyzed by SDS-PAGE followed by Coomassie blue staining. An aliquot of the input material was analyzed in the lower panels.

*trans*-interactions of Lnp molecules. Surprisingly, the P domain is not well conserved, even among higher eukaryotes, and is predicted to be unstructured. We therefore postulate that its folding is induced by the preceding CC2 domain, by interaction with a P domain from another Lnp molecule, or perhaps by an interaction with lipids.

The $Zn^{2+}$-finger does not seem to be absolutely essential for *trans*-interactions because stacked discs could still be observed with a $Zn^{2+}$-finger deletion mutant. On the other hand, Lnp fragments containing the $Zn^{2+}$-finger are dominant-negative when added before reconstitution, probably because these fragments interfere with *trans*-interactions of the P domains. Although not essential, the $Zn^{2+}$-finger domain may enhance *trans*-interactions of Lnp molecules by its ability to form dimers (16, 19).

We hypothesize that the observed *trans*-interactions of Lnp molecules have physiological significance. Lnp localizes preferentially to three-way junctions of the ER network, although it can also be found in tubules. We postulate that in metazoans, Lnp engages in *trans*-interactions when junctions come close to one another in a 3D network. Three-way junctions are small, triangular membrane sheets (14) and offer a better geometry than tubules for *trans*-interactions between multiple Lnp molecules. These *trans*-interactions may explain why Lnp preferentially localizes to tubular junctions and stabilizes them (13, 14, 15, 16). Although it is difficult to demonstrate *trans*-interacting Lnp molecules in vivo, the same mutations or dominant-negative constructs that inactivate Lnp in mammalian cells or *Xenopus* extracts cause the disruption of bicelle stacking in vitro.

Lnp-containing three-way junctions may not be able to undergo efficient membrane fusion, either because Lnp prevents ATL from entering the junctional sheets or because the membranes become less deformable. This model is consistent with the observation that overexpression of Lnp in mammalian tissue culture cells leads to expanded junctional sheets with ATL sitting at the edges and that co-overexpression of ATL restores a normal ER network (16). The postulated function of Lnp would be important at the center of mammalian cells, where the larger volume allows the ER network to be dense in all three directions. Although details of such a 3D network cannot be well visualized by normal light microscopy, Lnp-tethered membranes may account for clustered tubular junctions recently observed by super-resolution light microscopy (28). The prominence of ER sheets at the center of the cell might indicate that Lnp does not completely inhibit all ATL-mediated fusion into sheets. At the periphery of the cell, the network is essentially two dimensional, and Lnp would not be required to prevent excessive fusion. This model may explain why in mammalian cells lacking Lnp, sheets become more prominent and the number of junctions and tubules is diminished, with the residual network localized at the periphery of the cells (14, 15, 16). Our results can also explain the role of Lnp in the tubule-to-sheet conversion during mitosis; the inactivation of Lnp by phosphorylation of the P domain would abolish *trans*-interactions of Lnp molecules and allow membranes to undergo ATL-mediated fusion (16). Finally, the model can explain the formation of large membrane structures in mammalian cells when Lnp is expressed at very high levels (16). We speculate that these structures are actually Lnp-tethering membrane sheets, where the separation of individual sheets is below the resolution of light microscopy. These sheets would be very different from the ones generated by excessive membrane fusion in the absence of Lnp.

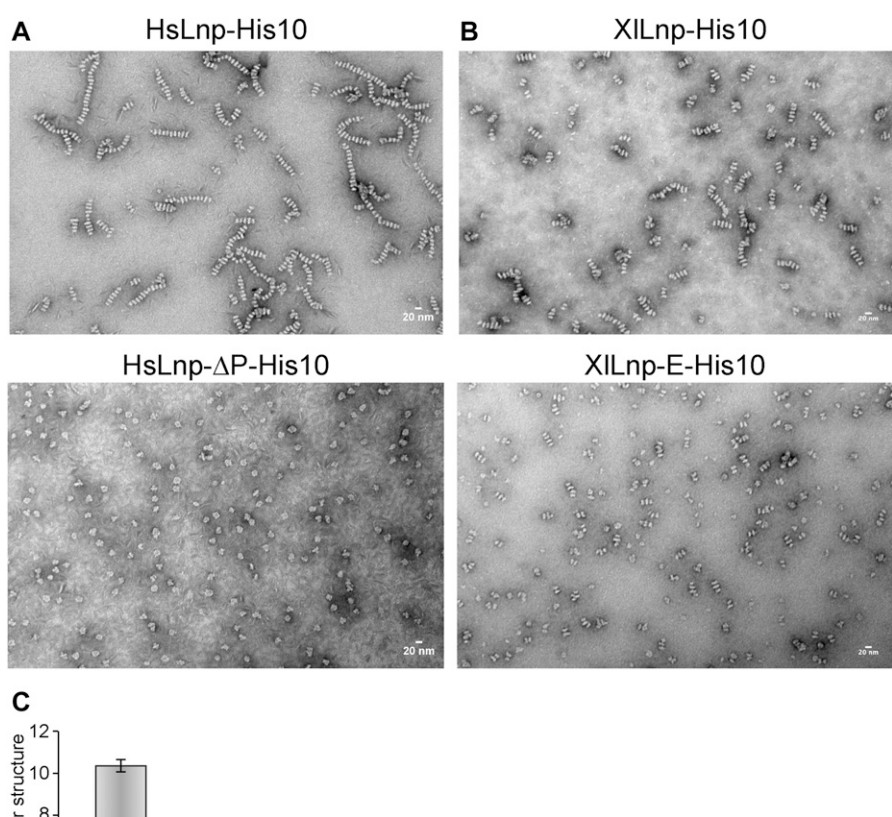

**Figure 7. Lnp-mediated stacked bicelle formation requires the P domain.**
**(A)** Purified full-length human Lnp or Lnp lacking the P domain (ΔL134–G234) was reconstituted with phospholipids at a 1:5 protein-to-lipid ratio and the samples were analyzed by negative-stain EM. Scale bar, 20 nm. **(B)** As in (A), but with full-length *Xenopus* Lnp or Lnp carrying phosphomimetic mutations in mitotic phosphorylation sites (16) (S177E, S179E, S188E, S192E, T198E, S206E, S215E, T219E, S222E, and S231E) in the P domain (XlLnp-E-His10). **(C)** Quantification of the average number of discs per stacked structure for samples shown in (A) and (B). The data plotted reflect the mean ± standard deviation from three independent experiments. About 100 stacked structures per sample were quantified in each independent experiment.

In our in vitro experiments, we reconstituted Lnp that lacks the N-terminal myristic acid. Previous in vivo experiments have shown that unmodified Lnp does not localize to three-way junctions and causes ER morphology defects (16, 18). The N-terminal myristic acid might be transiently embedded in the lipid bilayer. Because myristic acid is not a very strong membrane anchor (29), we speculate that it transitions in and out of the bilayer. When inserted into the bilayer, the helix formed by CC1 would probably lie flat on top of the membrane, where it might not be able to interact with CC2. When the myristic acid is not inserted, CC1 would interact with CC2 and form a vertical stalk pointing away from the membrane that would allow Lnp *trans*-interactions (Fig 3F). A reversible untethering of membranes might be required in vivo to allow dynamic changes in ER morphology. In our in vitro system, unmodified Lnp would be permanently locked in a conformation that can engage in *trans*-interactions.

Lnp is conserved in all eukaryotes, including yeast. Like metazoan Lnp, the *S. cerevisiae* protein contains two TM segments, two coiled-coil segments flanking them, and a Zn$^{2+}$-finger domain. However, yeast Lnp does not contain the myristoylation site at the N-terminus and it probably does not undergo mitotic phosphorylation, consistent with the absence of a tubule-to-sheet transition of the yeast ER during mitosis. We therefore speculate that yeast Lnp permanently prevents excessive membrane fusion by the GTPase Sey1p. Consistent with this assumption, Lnp and Sey1p have been reported to have antagonizing effects on ER morphology (13).

## Materials and Methods

### Plasmids

Full-length *Xenopus* and human *LNP* were cloned into the pET21b vector, adding a C-terminal His10 tag. Variants of *Xenopus* and human *LNP* were generated using site-directed mutagenesis. Full-length *Xenopus LNP* was also cloned into the pET21b vector, adding a C-terminal 3C protease cut site followed by a His10 tag. Human *LNP* with a C-terminal tobacco etch virus protease cut site and an SBP tag was cloned into the pRS425 vector containing a *GAL1*

promoter and a *CYC1* terminator for expression in *S. cerevisiae*. Constructs expressing variants of His6-tagged cytLnp were described previously (16).

## Protein expression and purification

All membrane proteins, with the exception of SBP-tagged human Lnp, were expressed in *E. coli* BL21-CodonPlus (DE3)-RIPL cells (Agilent). Expression was induced at $OD_{600}$ ~ 0.6 with 250 μM isopropyl-β-D-thiogalactopyranoside at 16°C for 18 h. SBP-tagged human Lnp was expressed in *S. cerevisiae* in the presence of galactose, as described previously (12). The purification of membrane proteins from *E. coli* and *S. cerevisiae* was performed essentially as described (12). Briefly, the cells were disrupted in lysis buffer (20 mM Tris, pH 7.5, 300 mM NaCl, 10% glycerol, 20 mM imidazole, 2 mM β-mercaptoethanol, 1 mM phenylmethylsulfonyl fluoride, and protease inhibitors). For SBP-tagged human Lnp, glycerol (which reduces the binding of SBP to streptavidin resin) and imidazole were omitted. Bacterial cells were broken by high-pressure homogenization in an M-110P microfluidizer (Microfluidics), whereas yeast cells were homogenized with a bead beater (BioSpec) with glass beads of 0.5-mm diameter (BioSpec). Cell debris and unbroken cells were removed by a low-speed spin and the clarified lysate was centrifuged at 100,000 *g* for 1 h to separate membranes from the cytosol. The membrane pellet was solubilized with lysis buffer supplemented with either 1% DDM for *Xenopus* Lnp or 1% Triton X-100 for human Lnp for 1 h at 4°C. Insoluble material was removed by centrifugation at 100,000 *g* for 1 h. The clarified membrane extract was incubated with Ni-NTA resin (Thermo Fisher Scientific) for His10-tagged proteins or streptavidin resin (Gold Biotechnology) for SBP-tagged proteins. His10-tagged *Xenopus* or human Lnp proteins were eluted with lysis buffer containing 250 mM imidazole in the presence of 0.03% DDM or 0.05% Triton X-100, respectively. SBP-tagged human Lnp was eluted from the streptavidin resin in the presence of 2 mM biotin. Lnp proteins were further purified by SEC on a Superose 6 column (GE Healthcare) pre-equilibrated with the SEC buffer (20 mM Hepes, pH 7.5, 150 mM KCl, and 1 mM dithiothreitol) containing the corresponding detergent. Fractions were pooled and concentrated, and the purity of the sample was analyzed by SDS-PAGE. Concentrations were determined based on $A_{280}$ reading for DDM-containing samples and a 660-nm protein assay (Thermo Scientific) for Triton X-100–containing samples. For purification of the untagged version of *Xenopus* Lnp, 3C protease was added for an on-column cleavage at 4°C overnight after a brief washing of the Ni-NTA resin. For purification of untagged human Lnp, the SBP tag was cleaved during an overnight on-column cleavage at 4°C in the presence of tobacco etch virus protease. In both cases, flow-through fractions were collected, further purified using Superose 6 columns, pooled, and concentrated. The purification of cytLnp and its variants was performed as previously described (16).

## Preparation of liposomes

Liposomes containing 60:33.4:6.6 mole percent of 1,2-dioleoyl-*sn*-glycero-3-phosphocholine:1,2-dioleoyl-*sn*-glycero-3-phosphoethanolamine:1,2-dioleoyl-*sn*-glycero-3-phosphoserine (Avanti Polar Lipids) were

prepared as described previously (12). Briefly, a thin lipid film was formed by drying chloroform–lipid mixtures under $N_2$ gas and vacuum. The lipid film was hydrated in SEC buffer to yield large multilamellar vesicles. These were subsequently subjected to 10 freeze–thaw cycles, followed by extrusion through a 100-nm pore size filter to yield large unilamellar vesicles.

## Reconstitution into liposomes

Proteins were inserted into liposomes using a detergent-mediated reconstitution method, as described previously (9, 12, 30, 31). Briefly, proteins and freshly extruded liposomes were mixed at the desired protein-to-lipid ratio in the presence of 0.1% detergent (DDM for *Xenopus* Lnp and Triton X-100 for human Lnp) for 30 min at room temperature. For example, if a protein-to-lipid molar ratio of 1:5, 1:10, 1:20, 1:40, or 1:200 was used, the lipid was added to a final concentration of 32 μM, 64 μM, 129 μM, 256 μM, or 1.29 mM, respectively. A small amount of Bio-Beads SM-2 Resin (Bio-Rad) was added to remove detergent and the mixture was incubated for 1 h at room temperature. This step was repeated three more times and the sample was then used for EM analysis or flotation.

## Negative-stain EM

Samples were diluted 10 times with SEC buffer and applied onto glow-discharged carbon-coated copper grids (Pelco, Ted Pella Inc.). After a 1-min incubation on the grid, excess sample was blotted off with filter paper. The grids were washed twice with water and incubated twice with 0.75% uranyl formate for 20 s. The grids were blown dry and imaged with a JEOL 1200EX transmission electron microscope operated at an acceleration voltage of 80 kV and equipped with a tungsten filament and an AMT 2kCCD camera. Images were captured at a magnification of 50,000.

## Cryo-ET sample preparation and imaging

Three microliters of reconstituted His10-tagged human Lnp was mixed 1:1 with 6-nm protein A-gold (Aurion) as fiducial markers and applied to glow-discharged R2/2 Cu 300-mesh holey carbon-coated support grids (Quantifoil). The grids were blotted using Whatman No. 41 filter paper for ~6 s in a humidified atmosphere and plunge-frozen in liquid ethane in a homemade device. The grids were maintained under liquid nitrogen and transferred to the electron microscope at liquid nitrogen temperature.

Tomograms were typically collected from +60° to −60° at tilt steps of 2° and −3 to −5 μm defocus using a Tecnai Polara (FEI) microscope, equipped with a field emission gun operating at 300 keV and a K2 Summit direct electron detector with a post-column energy filter (Gatan). Dose-fractionated data (8–10 frames per projection image) were collected at a nominal magnification of 61,000 (corresponding to a pixel size of 3.5 Å) using Digital Micrograph (Gatan). The total dose per tomogram was less than ~80 e/$Å^2$.

Images were aligned using the gold fiducial markers and contrast transfer function–corrected, and tomographic volumes were reconstructed by weighted back-projection using the IMOD software (Boulder Laboratory) (32). Contrast was enhanced by nonlinear anisotropic diffusion filtering in IMOD (33). To generate

3D–rendered views of the sample, subtomogram averaging of 115 discs was used. Two-point coordinates corresponding to the upper and lower edges of each disc were manually chosen and extracted from twice-binned tomograms in IMOD, with alignment and averaging carried out using particle estimation for electron tomography (34). Subtomogram averages were placed back into the tomographic volume for display using Chimera (UCSF).

### Calculations of bicelle diameter and spacing

Slices through tomograms were analyzed by drawing a plot profile of gray values in ImageJ (35), which could be exported as a function of distance. Grayscale values are calculated from the intensity of light value of each pixel in the image, with black values measured as the lowest and white values the highest.

### Flotation of proteoliposomes

An aliquot of the reconstituted sample was mixed with 80% Nycodenz predissolved in SEC buffer at 40% in a total volume of 100 μl. The sample was laid at the bottom of a 175-μl ultracentrifuge tube (Beckman, 342630). Then, the samples were overlaid with 50 μl of 30% Nycodenz, 50 μl of 15% Nycodenz, and 30 μl of SEC buffer to form a 0–40% Nycodenz step gradient. The sample was centrifuged in a swinging-bucket rotor at 100,000 $g$ for 1 h. Six fractions (F1–F6) were collected from the top to the bottom and analyzed by SDS-PAGE.

### Pull-down experiments

SBP-tagged human Lnp was incubated with either His10-tagged human Lnp at a molar ratio of 1:5 or Lnp lacking the $Zn^{2+}$-finger domain at a molar ratio of 1:3 or 1:5 for 1 h at 4°C in binding buffer (20 mM Hepes, pH 7.5, 100 mM KCl, 1 mM dithiothreitol, and 0.05% Triton X-100). One percent of the sample was saved as input material and the remainder was incubated with streptavidin resin for 1–2 h at 4°C, washed with binding buffer, and eluted with binding buffer supplemented with 2 mM biotin. The input and eluate fractions were analyzed by SDS-PAGE and Coomassie blue staining.

### Affinity purification of antibodies

Polyclonal antibodies to *Xenopus* Lnp, ATL, and Rtn4a were raised in rabbits using purified His6-tagged cytLnp, full-length ATL, and GST-tagged Rtn4a (positions 1,023–1,043) proteins as the antigens, respectively (6, 16). Antibodies were further affinity-purified as described previously (16). Briefly, the crude serum was incubated with the immunogen cross-linked onto Affigel-15 resin (Bio-Rad) and eluted with a low pH glycine/HCl buffer, and neutralized. The buffer was exchanged to 20 mM Hepes, pH 7.5, 150 mM KCl, and 250 mM sucrose. The concentration of purified antibodies was determined based on $A_{280}$ absorbance.

## Supplementary Information

## Acknowledgements

We thank Misha Kozlov for stimulating discussions, the EM facility at HMS for help, and Marco Catipovic and Misha Kozlov for critical reading of the manuscript. S Wang was supported by a fellowship from the Charles King Trust and RE Powers by a NIGMS T32 training grant (GM008313). We acknowledge the Max Planck Society and University of Exeter for supporting V Gold, in particular Werner Kühlbrandt and Deryck Mills at the Max Planck Institute of Biophysics. TA Rapoport is a Howard Hughes Medical Institute Investigator.

### Author Contributions

S Wang: Conceptualization, data curation, software, formal analysis, validation, investigation, visualization, methodology, writing—original draft, review, and editing.
RE Powers: Data curation, software, formal analysis, validation, investigation, visualization, methodology, writing—review and editing.
V Gold: Data curation, software, formal analysis, funding acquisition, validation, visualization, methodology, writing—review and editing.
TA Rapoport: Conceptualization, resources, formal analysis, supervision, funding acquisition, investigation, methodology, project administration, writing—original draft, review, and editing.

### Conflict of Interest Statement

The authors declare that they have no conflict of interest.

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
