## [Reviewer comments · Life Science Alliance]

The ER morphology-regulating lunapark protein induces the formation of stacked bilayer discs

Songyu Wang, Robert E. Powers, Vicki Gold, and Tom A. Rapoport

DOI: 10.26508/lisa.201700014

Review timeline:

Submission date:	17 December 2017
Editorial Decision:	19 December 2017
Revision received:	21 December 2017
Accepted:	21 December 2017

Report:

(Note: Letters and reports are not edited. The original formatting of letters and referee reports may not be reflected in this compilation.)

1st Editorial Decision

19 December 2017

Thank you for submitting your manuscript entitled "The ER morphology-regulating lunapark protein induces the formation of stacked bilayer discs" to Life Science Alliance. Your manuscript has been reviewed at another journal, and you provided a revised version and a point-by-point response to the previous round of review to us.

The three referees pointed out that the data provided are convincing and solid and the experiments well-controlled.

We appreciate the revision and the way you addressed the referees' concerns regarding lack of in vivo support of your conclusions and the proposed reasoning for the observed stacked discs that are bicelle-like. We are thus happy to accept your manuscript in principle for publication in Life Science Alliance.

Congratulations on this very nice work!

Before sending you the official acceptance letter, a few editorial points need to be addressed:

- please provide your ORCID (you should get an auto-link to fill in this information)
- please add callouts in your text for figures S1B, S4A, S4C, S4D, S4E
- please add the scale bar length to the figure legends
- please indicate in the figure legends that the images in Figure 1B and Figure 5B are the same.
- please fill in the license form (you should receive an auto-link)

Thank you very much, I am very much looking forward to publishing your work!

2nd Editorial Decision

21 December 2017

Thank you for contributing your Research Article entitled "The ER morphology-regulating lunapark protein induces the formation of stacked bilayer discs". It is a pleasure to let you know that your manuscript is now accepted for publication in Life Science Alliance. Congratulations on this interesting work.

Your manuscript will now progress through copyediting and proofing.

DISTRIBUTION OF MATERIALS:

Again, congratulations on a very nice paper. I hope you found the review process to be constructive and are pleased with how the manuscript was handled editorially. We look forward to future exciting submissions from your lab.